# Excited-State Intramolecular Proton Transfer Dyes with Dual-State Emission Properties: Concept, Examples and Applications

**DOI:** 10.3390/molecules27082443

**Published:** 2022-04-10

**Authors:** Timothée Stoerkler, Thibault Pariat, Adèle D. Laurent, Denis Jacquemin, Gilles Ulrich, Julien Massue

**Affiliations:** 1Institut de Chimie et Procédés pour l’Energie, l’Environnement et la Santé (ICPEES), Equipe Chimie Organique pour la Biologie, les Matériaux et l’Optique (COMBO), UMR CNRS 7515, Ecole Européenne de Chimie, Polymères et Matériaux (ECPM), Université de Strasbourg, 25 Rue Becquerel, CEDEX 02, 67087 Strasbourg, France; stoerkler@etu.unistra.fr (T.S.); thibault.split@gmail.com (T.P.); gulrich@unistra.fr (G.U.); 2Chimie et Interdisciplinarités: Synthèse, Analyse et Modélisation (CEISAM), UMR CNRS 6230, Nantes University, 44322 Nantes, France; adele.laurent@univ-nantes.fr

**Keywords:** fluorophores, ESIPT fluorescence, dual-state emission, ab initio calculations

## Abstract

Dual-state emissive (DSE) fluorophores are organic dyes displaying fluorescence emission both in dilute and concentrated solution and in the solid-state, as amorphous, single crystal, polycrystalline samples or thin films. This comes in contrast to the vast majority of organic fluorescent dyes which typically show intense fluorescence in solution but are quenched in concentrated media and in the solid-state owing to π-stacking interactions; a well-known phenomenon called aggregation-caused quenching (ACQ). On the contrary, molecular rotors with a significant number of free rotations have been engineered to show quenched emission in solution but strong fluorescence in the aggregated-state thanks to restriction of the intramolecular motions. This is the concept of aggregation-induced emission (AIE). DSE fluorophores have been far less explored despite the fact that they are at the crossroad of ACQ and AIE phenomena and allow targeting applications both in solution (bio-conjugation, sensing, imaging) and solid-state (organic electronics, data encryption, lasing, luminescent displays). Excited-State Intramolecular Proton Transfer (ESIPT) fluorescence is particularly suitable to engineer DSE dyes. Indeed, ESIPT fluorescence, which relies on a phototautomerism between normal and tautomeric species, is characterized by a strong emission in the solid-state along with a large Stokes’ shift, an enhanced photostability and a strong sensitivity to the close environment, a feature prone to be used in bio-sensing. A drawback that needs to be overcome is their weak emission intensity in solution, owing to detrimental molecular motions in the excited-state. Several strategies have been proposed in that regard. In the past few years, a growing number of examples of DSE-ESIPT dyes have indeed emerged in the literature, enriching the database of such attractive dyes. This review aims at a brief but concise overview on the exploitation of ESIPT luminescence for the optimization of DSE dyes properties. In that perspective, a synergistic approach between organic synthesis, fluorescence spectroscopy and ab initio calculations has proven to be an efficient tool for the construction and optimization of DSE-ESIPT fluorophores.

## 1. Introduction

In the current context of sustainable development, there is a great deal of research aimed at developing purely organic luminescent materials, that is π-conjugated organic scaffolds capable of absorbing light at given wavelengths and reemitting it as lower-energy photon through a radiative deactivation following various photophysical processes. Expedite syntheses preferentially from biosourced or bioinspired starting materials, chemical and photochemical stability, biocompatibility and modular photophysical properties are among the major goals, not to say requirements, which need to be fulfilled prior to embedment in matrixes or devices for practical applications [1,2,3]. On top of preserving limited natural resources like metals or rare earths, the search for original organic-based luminescent scaffolds is fueled by the advantages of these hydrocarbon structures: low-cost, processability, stability and solubility, and a fluorescence emission which can be fine-tuned by small structural changes. Additionally, their capacity to respond to external stimuli, i.e., the presence of selected substrates or changes in the physical properties of their environment (temperature, pressure, polarity, pH, viscosity…) by adjusting their optical signatures is also scrutinized [4,5]. In the context of designing luminescent molecular skeletons, organic synthesis and first-principle modelling (typically performed with time-dependent density functional theory, TD-DFT) appear as complementary but intertwined scientific tools. The potential applications of organic fluorescent chromophores can be roughly divided between those requiring fluorophores fully dissolved in a given medium, organic or aqueous (emission in solution) and those where fluorescence must be observed in their solid forms, as amorphous powders, crystals, thin films or various matrixes. Organic fluorophores displaying emission in solution, i.e., fully solvated, are typically used as fluorescent tools or molecular beacons for biomedical applications, such as imaging of cellular organelles (lysosomes, endoplasmic reticulum, Golgi apparatus, nucleus…), cells or tissues, [6] selective imaging of tumors, [7] cellular mapping, bioconjugation to macromolecules (proteins, nucleic acids) [8] and so on. Solution-emitting luminescent dyes can be also applied in materials for selective sensing, by modification of their photophysical properties upon recognition of substrates of interest [9]. These probes can provide information about the concentration of targets in organic, as well as in biological media. The most promising sensors involve ratiometric processes which can correlate accurate concentration and ratios of emission bands [10].

Solid-state emitting organic fluorophores have found complementary applications, mainly as active layers embedded in devices, such as organic light-emitting diodes (OLED), organic field-effect transistors (OFET), light-emitting electrochemical cells or in organic photovoltaic (OPV) [11,12,13]. Other applications of solid-state emissive dyes include lasing, data encryption (anti-counterfeiting security inks, logic gates), solid-state sensors and any applications where organic dyes are to be confined in a matrix at high local concentrations [14,15,16,17].

The vast majority of organic dyes display intense fluorescence emission, only as solvated in dilute solutions owing to strong radiative deactivations of their excited states. Indeed, most π-conjugated organic materials present a strong degree of planarity and rigidity, leading to aggregation in the solid-state enabling powerful non-radiative deactivation channels. This aggregation-caused quenching (ACQ) process is observed in many π-conjugated hydrocarbon structures (coumarins, cyanines, polycyclic aromatic hydrocarbons such as pyrene, anthracene and perylene, fused heterocycles and so on, see Figure 1). It is nevertheless worth noting that some of the listed dyes such as pyrene, can also present intense fluorescence stemming from the formation of excimers [18,19,20]. These detrimental effects limited for a long time the practical applications of organic luminophores in films or devices. To circumvent ACQ, Tang et al. popularized the concept of aggregation-induced emission (AIE), which consists of molecular rotors having in their structure multiple σ bonds, at the origin of numerous molecular motions in dilute solution favoring non-radiative deactivations. Therefore, in contrast to the majority of organic structures described before, AIE dyes present strong fluorescence quenching in dilute solution, whereas in their aggregated/solid-state, the restriction of molecular motions leads to the appearance of a strong luminescence (Figure 1) [21,22,23]. Over the years, many molecular structures, mainly composed of arenes or heterocycles, have been described as possessing AIE properties. Among these compounds, tetraphenylethene (TPE) derivatives are among the most reported but one can also quote examples derived from tetraphenylpyrazine (TPP), tetraphenyl-1,4-butadiene (TPBD), distyreneanthracene (DSA) or quinoline-malononitrile (QM) as chromophores displaying AIE [24].

Dual-state emission (DSE) molecules correspond to luminescent molecular structures which show intense fluorescence emission both in solution and in the solid state. These dyes have been far less studied but show nevertheless a great interest to the community since they can simultaneously act as solution- and solid-state emitters and therefore target a wider range of applications. This field of research still being in its infancy, there are no clear guidelines for the elaboration of such attractive dyes yet, although their optical properties observed in multiple media seem to arise from a tenuous balance between quasi-planarity, semi-rigidity and solubility. The presence of heteroatoms in the π-conjugated scaffold of DSE dyes is also often observed, presumably inducing intramolecular charge transfer (ICT) within the structures, creating dipole moments in the excited-state and promoting significant geometrical reorganization between the ground and excited states.

Different scaffolds have been reported to show DSE properties, among which one can quote benzimidazoles, [25] triphenylamine-based compounds, [26] phtalamides, [27] benzo [1,2,5]thiadiazole, [28] naphtho [2,1-b]benzofurans, [29] boron complexes, [30,31] diphenylpyrrole [32] or even simple benzene rings rationally substituted by donors and acceptors [33]. The range of π-conjugated structures displaying DSE, along with the photophysical mechanisms driving DSE properties and their applications, has been recently analyzed by Rodríguez-Molina et al., in two extensive review articles [34,35].

Fluorescent dyes presenting a molecular backbone undergoing Excited-State Intramolecular Proton Transfer (ESIPT) emission appear particularly attractive in the context of DSE dyes engineering. Indeed, ESIPT corresponds to a photoinduced tautomerization between an excited normal (N*) and a tautomeric species (T*), therefore inducing significant dynamics in the excited-state, which is beneficial to prevent strong π-stacking processes (Figure 2) [36,37,38,39]. In some selected cases of solvation or upon electronic effects, a dual N*/T* emission can be observed [40]. ESIPT luminescence is typically observed in heterocycles presenting an intramolecular hydrogen bond forming five-, six- or seven-membered rings in the ground-state and involves in the majority of cases phenol (E*) and keto (K*) tautomers. Many beneficial characteristics arise from these specific excited-state dynamics, notably large Stokes shifts and intense solid-state emission [41,42,43]. ESIPT emitters have found numerous applications in the fields of optoelectronics, sensing, imaging, along with various luminescent displays [44,45,46]. Over the years, a large variety of molecular backbones showing ESIPT emission has been reported, with a common drawback being the weak emission intensity in dilute solution due to detrimental molecular motions in the excited-state, opening highly effective non-radiative deactivations channels [47,48,49,50], notably through an accessible conical intersection (CI) corresponding to the twisting of the interesting double bond of the keto form.

## 2. Scope of This Short Review/Perspective

In the past few years, publications have started to tackle the problem of weak ESIPT solution-state emission through various synthetic tricks and strategies [51] and a growing number of ESIPT dyes display fluorescence in solution, while keeping strong emission in the solid-state, a hallmark of ESIPT luminophores. The aim of this short review is to enlighten the reader on the (so far) limited but reducing non-radiative desexcitations displaying intense fluorescence emission intensity in multiple environments. To shorten the range of examples and to select only the most promising ESIPT/DSE dyes, only fluorophores showing at least quantum yields (QY) of 10% both in solution and solid media are reported in the present account. Examples of ESIPT/DSE emitters will be first presented followed by a summary of their optical properties in various states in solvents of different polarities, powders, crystals, films or other solid-state matrixes depending on studies (Table 1), some brief TD-DFT considerations and possible future developments.

## 3. Examples of ESIPT/DSE Emitters

This section summarizes the various π-conjugated structures showing ESIPT/DSE properties which will be tentatively classified by the nature of the main π-conjugated scaffold.

### 3.1. 2-(2′-Hydroxyphenylbenzazole) (HBX) Fluorophores

Among ESIPT dyes, 2-(2′-hydroxyphenyl)benzazole (HBX) fluorophores which are composed of a benzazole ring connected to a phenol at the *ortho* position, have been often studied as model dyes due to their facile synthesis, stability, biocompatibility and possibility to adjust their photophysical properties with small structural inputs [52,53,54]. For example, their constitutive heterocycle can be altered to provide 2-(2′-hydroxyphenyl)benzoxazole (HBO), 2-(2′-hydroxyphenyl)benzothiazole (HBT) or 2-(2′-hydroxyphenyl)benzimidazole (HBI) derivatives. HBT dyes display the most redshifted emission due to the larger polarizability of the thiazole ring [55] whereas HBI derivatives show the highest QY, thanks to decreased non-radiative deactivations [56]. The impact of incorporating electrodonors or acceptors leads to a large panel of effects, such as the emergence of an ICT emission, increase of solution QY, stabilization of (possibly emissive) anionic species or frustration of the ESIPT process [57,58,59]. HBX dyes presenting DSE properties have also been reported in recent literature. The key parameter to combine ESIPT and DSE is to enhance radiative deactivation in solution to increase fluorescence intensity while conserving a valuable QY in the solid-state. Increasing molecular rigidity on the main molecular backbone is usually a good strategy to increase solution-state fluorescence by reducing molecular motions and notably making the access to the above-mentioned CI energetically more difficult. This explains why the concept of restricted access to conical intersection (RACI) has been proposed to rationalize the reduced non-radiative deactivation in solution [60]. As stated above, HBX can be divided in three groups, along with the nature of the heteroatom, i.e., nitrogen, oxygen or sulfur; each of those providing access to HBI, HBO and HBT dyes. Examples of DSE dyes for these three classes of dyes are presented in Figure 3, Figure 4 and Figure 5.

**Figure 3 molecules-27-02443-f003:**
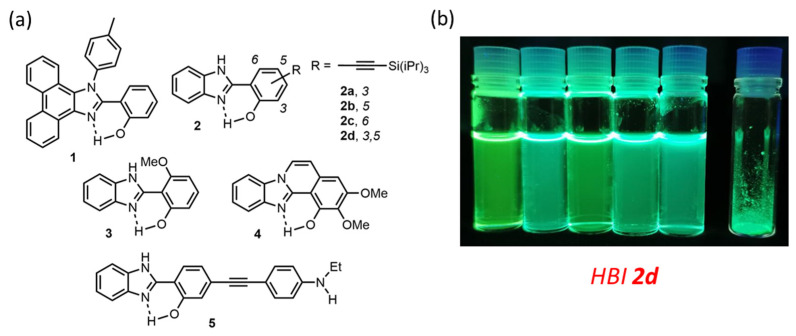
(**a**) Examples of HBI dyes showing DSE properties [61,62,63,64,65] and (**b**) Photographs of 3,5-tri(isopropyl)silyl ethynyl-extended HBI dye **2d** under irradiation in solution (toluene, ethanol, THF, acetonitrile and DMF) and as powders (λ_exc_ = 365 nm) [62]. Adapted from ref [62] Copyright 2021 John Wiley and Sons.

Incorporating the benzimidazole ring seemed particularly attractive since several studies hinted that HBI dyes present significantly larger fluorescence QY in solution, as compared to their oxygen or sulfur analogs, by reducing non-radiative desexcitations in the excited-state [66]. In addition, for HBI dyes, several examples have been reported where the substitution leading to enhanced molecular rigidity clearly leads to an increase of fluorescence intensity in solution (Figure 3). We also underline the marked unsymmetrical character of these derivatives, which limits the possibility of π-stacking, a feature beneficial for maintaining bright emission in the solid-state. Notably, N-arylated 9,10-phenanthroimidazole derivative **1**, a rigidified analog of HBI, was one of the earliest examples showing ESIPT/DSE properties [61]. Following the strategy of rigidification-induced DSE engineering, Pariat et al., described a series of ethynyl-extended HBI dyes **2a**–**2d** which pinpointed a significant influence of the position of functionalization on the resulting QY in toluene and in potassium bromide pellets [60] Takagi et al., disclosed an original strategy to enhance solution fluorescence intensity by introducing a methoxy group at the 6-position of the phenol and they engineered the supramolecular H-bonded rigidification with the benzimidazole ring, as in derivative **3** [63]. They also investigated the influence of structural rigidification by ring fusion in HBI **4** and found out that the presence of the ethylene bridge triggered not only an increase of fluorescence intensity but also the enhancement of the ESIPT process, as evidenced by the main observation of the excited tautomer species K* [64]. Finally, a recent example by Munch et al., involved a π-extended mono N-alkylated ethynyl aniline HBI derivative **5** [65].

**Figure 4 molecules-27-02443-f004:**
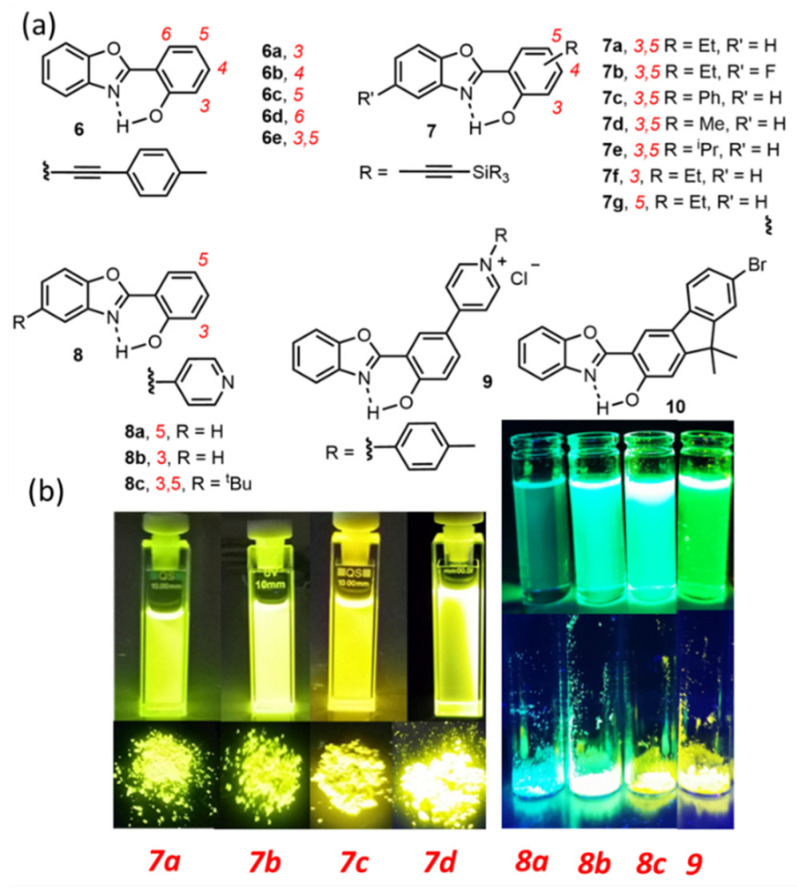
(**a**) Examples of HBO dyes showing DSE properties [67,68,69,70,71,72] and (**b**) Photographs of HBO dyes **7a–d** and **8a–c**/**9** under irradiation in solution (toluene or dichloromethane) and as powders (λ_exc_ = 365 nm) [69,71]. Adapted from Ref. [69] Copyright 2019 Elsevier. Adapted from Ref. [71] Copyright 2021 American Chemical Society.

**Figure 5 molecules-27-02443-f005:**
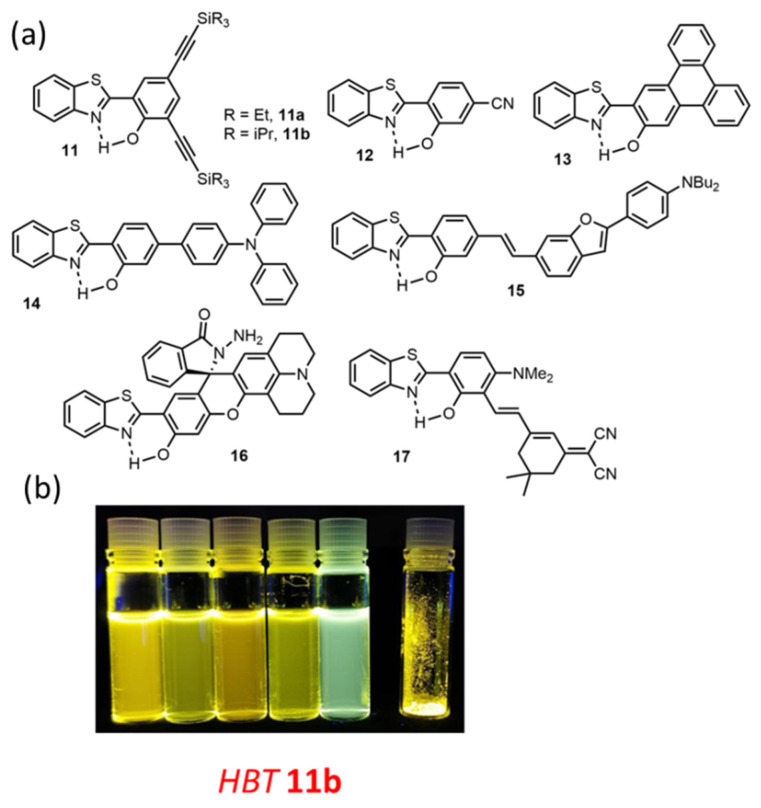
(**a**) Examples of HBT dyes showing DSE properties [62,69,73,74,75,76,77,78]. and (**b**) Photographs of HBT dye **11b** under irradiation in solution (toluene, ethanol, THF, acetonitrile and DMF) and as powders (λ_exc_ = 365 nm) [62]. Adapted from ref [62] Copyright 2021 John Wiley and Sons.

Similar features have been reported in the HBO series where some ESIPT/DSE probes have been recently described (Figure 4). Indeed, the insertion of ethynyl-extended tolyl or triaryl-/trialkyl-silyl substitution has significantly improved the fluorescence intensity in solution, in aprotic as well as polar protic environment [67,68,69,70]. The solution QY of the HBO dyes series **6** and **7** was increased up to twenty times, as compared to unsubstituted HBO analogs, while maintaining intense emission in the solid-state (Table 1). Another attractive strategy to engineer DSE features among ESIPT fluorophores is to stabilize excited tautomers through electronic delocalization [71]. HBO dyes **8**, substituted at the 5-position of the phenol by a pyridine ring displays strong blue fluorescence, which is further enhanced by protonation of the heteroring. TD-DFT calculations revealed that the corresponding pyridinium was able to stabilize the K* state by resonance and therefore disfavor non-radiative deactivations, leading to intense fluorescence in solution. The corresponding methyl pyridinium moieties were synthesized, showing full ESIPT process and leading to strong fluorescence in solution but also to a strong quenching of emission in the solid-state, presumably owing to unfavorable electrostatic interactions. Solid-state emission can be, however, beneficially recovered by substituting the methyl group for a tolyl, as in HBO **9**. Another example, HBO **10**, reported by Göbel et al., consists in the fusion of a fluorene moiety to the phenol ring, whose stiffness leads to dual-emission in various conditions [72].

HBT dyes have been far more studied than HBI or HBO analogs thanks to their red-shifted emission, reaching in some cases the red region of the visible spectrum, therefore making them more suitable for biological applications. The engineering of DSE dyes based on HBT appears therefore as a more challenging objective (Figure 5). Following the same approaches as for the other systems, molecular rigidification appears to be a good strategy to enhance solution-state emission of HBT dyes, even though the QY usually remains lower than for their HBO/HBI analogs (Table 1). The insertion of ethynyl-extended spacer at the 3,5 positions of the phenol ring, as in HBT **11**, led a drastic enhancement of emission intensity, regardless of the nature of the alkyl chain on the silyl groups [62,69]. Another interesting example of DSE emission lies in HBT **12** where the introduction of a strong electrowithdrawing such as a cyano group at the 4-position of the phenol ring appeared to be a key input to trigger strong solution-state emission, while keeping full ESIPT process [73]. Again, the theory of RACI, hypothesizing a least accessible conical intersection, thereby avoiding detrimental non-radiative deactivations, might be useful to explain these optimized properties. By extending the conjugation with a triphenyl moiety, HBT dye **13** exhibits bright red solid-state fluorescence with a large Stokes shift, a trademark of ESIPT derivatives [74]. Extension of π-conjugation as in HBT **14** and **15**, through an aryl or a styryl spacer triggers a strong frustration to the ESIPT process which can be full or partial, leading to ICT-based emission with strong fluorescence intensity in solution [75,76]. It is worth noting that a partial restoration of ESIPT is observed in the solid-state for **14** [75], and upon protonation in the case of HBT **15** [76], highlighting the small differences between the relative energies of the two excited species E* and K*. HBT **16** was conceived as a dual-channel fluorescent probe for the logic-based visualization of aging biomarkers (thiophenol and Hypochlorous acid HOCl) [77]. Finally, Kaur et al. designed HBT **17**, bearing a delocalized styryl spacer at the *ortho* position of the phenol, to tune the emission color [78].

### 3.2. Other Fluorophores

While HBX derivatives appear to be the leading DSE fluorophores with ESIPT properties reported up to date with several molecular engineering studies, others families of dyes have also been reported for their intense fluorescence emission in multiple environments. Five-membered oxazoline rings, which are structurally similar to benzoxazole yet more flexible, have been recently described as potential candidates for DSE emission (Figure 6) [72,79,80,81].

Göbel et al. studied the influence of electronic substitution in 2-(oxazolinyl)-phenol derivatives **18a–b**, as minimalistic ESIPT fluorophores. In particular, the introduction of strong electrowithdrawing substituents (CF_3_, CO_2_Me) at the *para* position of the oxazoline triggered intense fluorescence, as compared to other substituents or other substitution positions on the π-conjugated scaffold [79]. The same group also reported nitrile-substituted 2-(oxazolinyl)-phenols dyes **19a–d** where subtle modifications, e.g., the position of the nitrile substituents induced different effects [80]. Although all dyes displayed significant emission intensity in both solution and solid, dyes **19a–b** showed aggregation-induced emission enhancement (AIEE) behavior, while **19d** featured ACQ characteristics. The introduction of a fused fluorene moiety of the proton donor side enhanced both planarity and rigidity of the resulting dye **20** which in turn displayed reduced non-radiative deactivations [72]. Finally, Huang et al., reported the introduction of strong electrodonating (triphenylamine) or electrowithdrawing (triphenylboron) groups which led to enhanced ICT effects in the resulting dye **21** and a subtle modulation of the nature of the excited species (E*, K* or a dual emission E*/K*), emphasizing again the close relative energies of the excited states of the two tautomers [81].

A recent example of random scaffolds showing DSE properties, reported by Stoerkler et al., involved salicylaldehyde derivatives **23a–d** with push–pull structures, which can be easily synthesized in two steps only (Figure 7) [82]. Within this series of dyes, the connecting spacer between the ESIPT center (salicylaldehyde as acceptor) and the donor, i.e., ethynyl (**23a,b**), styryl (**23c**) or aryl (**23d**) was found to have a major influence on the photophysical properties (nature of the excited states, maximum emission wavelength, QY). Only the dyes with a strong donor such as amino groups displayed DSE properties with a finely tuned emission color in solution and in the solid-state [82].

Miscellaneous organic scaffolds showing ESIPT emission with DSE properties can be also found in the recent literature (Figure 8). Among these dyes, imidazolo [5,4-*b*]thieno [3,2-*e*]pyridine moieties as proton acceptors have been described with hydroxy or methanesulfonamide as proton donors [83]. The good DSE properties of dyes **24** and **25** were rationalized by a finely controlled self-assembly and a restriction of twisted intramolecular charge-transfer (TICT) processes. Another example is dye **26**, presenting a 2,2′-bipyridine-3,3′-diol-5,5′-dicarboxylic acid ethyl ester featuring two intramolecular hydrogen bonds [84]. One of the H-bonded six-membered rings is involved in the ESIPT emission while the other one participates in the rigidification of the overall molecular structure. The electroluminescence performance of this dye was evaluated in an OLED device.

A step forward in the development of ESIPT dyes with DSE properties was achieved with the red/NIR emission observed by the squaraine dyes series **27a–d** [85]. Guided by TD-DFT calculations, the authors demonstrate that the removal a simple phenyl ring in TPE-fused squaraine dyes was a valuable strategy to generate intense red/NIR emissions in solution and in crystals. An alleviation of the TICT mechanism within the structure of these dyes appears to be at the origin of these attractive photophysical properties. Moreover, due to their highly biocompatible emission wavelength, the squaraine series **27** was successfully employed for cell bioimaging. A final example involves 2,5-disubstituted-1,3,4-oxadiazoles **28** which shows an emission in the green region [86].

Other examples which fall into the scope of ESIPT dyes with DSE properties concern fluorophores with AIEE behavior. These dyes typically present significant fluorescence emission as molecular entities, fully dissolved in a solvent, although they possess structural features which make them prone to aggregation. Just like AIE dyes, aggregation leads to a strong enhancement of emission intensity. These dyes are also usually emissive in the solid-state where the molecular motions are hindered. In this category of DSE dyes, one can quote HBX and bowl-shaped tris(2-hydroxyphenyl)triazasumanene examples [87,88]. We would also like to emphasize an important class of ESIPT dyes, namely 3-hydroxyflavone which display tunable fluorescence properties in solution. Their emission in the solid-state is often mentioned, rarely measured but they undeniably belong to the DSE/ESIPT class of compounds [89,90,91].

## 4. Photophysical Properties

The key photophysical properties in solution and in the solid-state of all the dyes described in this account are summarized in Table 1.

**Table 1 molecules-27-02443-t001:** Photophysical data for all DSE/ESIPT dyes in solution and solid-state.

Dye	λ_abs_ (Sol.) (nm)	λ_em_ (Sol.) (nm)	Φ_f_ (Sol.)	Solv.	λ_em_ (Solid) (nm)	Φ_f_ (Solid)	Matrix	Ref
**1**	363	472	0.15	CH_2_Cl_2_	473	0.12	KBr	[62]
**2a**	353	496	0.51	toluene	470	0.36	KBr	[62]
**2b**	347	482	0.54	toluene	460	0.39	KBr	[62]
**2c**	346	479	0.49	toluene	488	0.68	KBr	[62]
**2d**	368	507	0.53	toluene	490	0.30	KBr	[62]
**3**	330	458	0.31	THF	460	0.16	powder	[63]
**4**	328	486	0.20	THF	484	0.16	crystal	[64]
**5**	366	414/477	0.19	benzene	470	0.13	KBr	[65]
**6a**	349	397/514	0.10	toluene	530	0.51	KBr	[67]
**6b**	349	489	0.19	toluene	504	0.63	KBr	[67]
**6c**	347	550	0.32	Toluene	504	0.60	KBr	[67]
**6d**	332	519	0.23	toluene	503	0.68	KBr	[67]
**6e**	368	550	0.30	toluene	547	0.48	KBr	[67]
**7a**	371	537	0.38	toluene	527	0.76	KBr	[68]
**7b**	373	551	0.43	toluene	532	0.61	KBr	[69]
**7c**	370	530	0.52	toluene	526	0.58	KBr	[69]
**7d**	368	539	0.49	toluene	534	0.70	KBr	[70]
**7e**	371	535	0.32	toluene	530	0.82	KBr	[70]
**7f**	340	538	0.28	toluene	514	0.53	KBr	[70]
**7g**	345	513	0.11	toluene	504	0.66	KBr	[70]
**8a**	332	497	0.12	CH_2_Cl_2_	496	0.38	KBr	[71]
**8b**	335	518	0.40	CH_2_Cl_2_	505	0.38	KBr	[71]
**8c**	347	520	0.58	CH_2_Cl_2_	541	0.22	KBr	[71]
**9**	372	543	0.50	CH_2_Cl_2_	563	0.29	KBr	[71]
**10**	397	452/520	0.37	CHCl_3_	520	0.34	powder	[72]
**11a**	378	570	0.22	toluene	558	0.52	KBr	[69]
**11b**	378	490/582	0.15	toluene	574	0.48	KBr	[62]
**12**	362	520	0.49	CH_2_Cl_2_	528	0.57	5-CB	[73]
**13**	400	605	0.08	toluene	605	0.23	Crystal	[74]
**14**	376	444	0.65	toluene	465/527	0.22	Film	[75]
**15**	424	521	0.87	toluene	573	0.19	KBr	[76]
**16**	350	534	0.12	PBS	534	0.51	powder	[77]
**17**	400	600	0.34	CH_2_Cl_2_	695	0.34	powder	[78]
**18a**	307	466	0.38	CH_2_Cl_2_	466	0.57	powder	[79]
**18b**	325	479	0.63	CH_2_Cl_2_	477	0.74	powder	[79]
**19a**	322	471	0.25	CH_2_Cl_2_	491	0.87	powder	[80]
**19b**	303	452	0.25	CH_2_Cl_2_	463	0.68	powder	[80]
**19c**	326	468	0.56	CH_2_Cl_2_	494	0.74	powder	[80]
**19d**	321	457	0.53	CH_2_Cl_2_	473	0.15	powder	[80]
**20**	450	481	0.17	CHCl_3_	481	0.38	powder	[72]
**21**	370	425	0.47	toluene	530	0.31	PS	[81]
**22**	368	433	0.60	toluene	530	0.55	PS	[81]
**23a**	407	490	0.68	toluene	467	0.88	PMMA	[82]
**23b**	388	506	0.29	toluene	473	0.44	PMMA	[82]
**23c**	436	593	0.64	CH_2_Cl_2_	528	0.83	PMMA	[82]
**23d**	396	532	0.56	CH_2_Cl_2_	482	0.45	PMMA	[82]
**24a**	366	520	0.48	benzene	535	0.11	powder	[83]
**24b**	366	523	0.74	benzene	540	0.39	powder	[83]
**24c**	380	512	0.69	benzene	523	0.42	powder	[83]
**24d**	369	505	0.85	benzene	515	0.53	powder	[83]
**24e**	362	510	0.51	benzene	495	0.25	powder	[83]
**25**	366	542	0.52	benzene	535	0.13	powder	[83]
**26**	388	521	0.75	CH_2_Cl_2_	530	0.51	crystal	[84]
**27a**	508/544	558/593	0.11	toluene	675	0.08	crystal	[85]
**27b**	532/560	579/616	0.27	toluene	656	0.73	crystal	[85]
**27c**	537/568	586/622	0.58	toluene	670	0.51	crystal	[85]
**27d**	536/568	582/621	0.43	toluene	682	0.41	crystal	[85]
**28**	343	500/535	0.43	CH_2_Cl_2_	544	0.54	powder	[86]

## 5. First-Principle Modelling

As ESIPT relies on a subtle energetic balance between two tautomers in their excited states, and as this balance is highly medium-dependent, it is no surprise that theoretical tools are used in the majority of the recent works to probe the underlying driving mechanisms guiding emission in ESIPT dyes. The ab initio methods applied to model the properties of ESIPT dyes have recently been reviewed in this journal [39], and we refer the interested reader to that very nice account for details about the static and dynamic methods that are available. We also do not intend to review all ESIPT theoretical studies published to date, but rather to give a flavor of the modelling in both solution and solid states, especially those discussing the non-radiative pathways in ESIPT dyes. Importantly, for historical reasons, the modelling of ESIPT was mainly done in gas phase and next in solution, so that studies tackling actual DSE fluorophores remain very scare.

To our knowledge, the first theoretical studies demonstrating that an easily accessible CI is close to the tautomeric excited-state minimum in the simplest ESIPT dyes were performed by the Robb group [92,93] with a multi-reference approach well-suited to obtain accurate CI estimates. Another seminal study discussing the importance of the deactivation pathway by twisting is due to Tsai et al., who disclosed the presence of accessible TICT-like forms on the potential energy surface of two HBI dyes using TD-DFT [94]. Several studies obtained similar conclusions for a variety of ESIPT systems, confirming that twisting should be limited in solution to obtain bright fluorescence, and therefore to allow the design of DSE dyes [95,96,97,98]. Let us now turn to the modelling of solvent effects, that are most of the time accounted for using continuum models, which advantageously allow neglecting the atomic structures of the solvent molecules. First, let us briefly evoke a specific methodological aspect. In most ESIPT compounds, the main characteristics of the E* and K* forms strongly differ. The former typically presents a larger oscillator strength and a larger excited-state dipole than the latter. This means that it is not straightforward to have a solvent model that is adequate for both tautomeric species, at least at the TD-DFT level. Indeed, one needs to consider that both the so-called linear-response and state-specific solvation contributions are larger in the enol than keto structure, and one should ideally account for both effects to obtain an accurate description of both species [99,100]. This can be achieved with the so-called cLR^2^ model proposed by Guido et al., that has been successfully applied to some challenging ESIPT dyes [101,102]. Second, we wish to point out that several studies used a PCM-TD-DFT approach to assess the correlation between the computed barrier for the twist separating the K* minimum to the twisted CI, and the experimentally measured quantum yield of emission [62,67,69]. For instance, it was shown that the relative measured quantum yields of **7f** (0.28) and **7g** (0.11) are directly related to a higher barrier to the CI for the former (0.12 eV) than the latter (0.06 eV), therefore confirming the RACI analysis [70]. Similarly, a cLR^2^ study of the barriers in the HBO, HBI, and HBT analogues **7e**, **2d**, and **11b**, led the authors to conclude that the computed barrier heights are roughly proportional to the observed quantum yields [62]. More refined studies in which the solvent dynamics is explicitly investigated through QM/MM approaches are also available accounting for polarization [103,104]. Although, one should certainly be cautious in applying TD-DFT for such twisting case or generalizing these works to other classes of fluorophores, these examples illustrate that there is hope in estimating the emission yields using readily accessible with theoretical data. Theoretical studies of the AIE effect for ESIPT dyes in the solid state have been mainly but not exclusively achieved for compact derivatives (salicylaldehyde, chalcone, quinazoline…), that do not strictly fall in the DSE category as the emission is typically very weak in solution [105,106,107,108,109,110,111]. Again, we wish to start here with a methodological note: adequate theoretical schemes have to be set up to allow accurate estimates of the impact of the environment on the computed emission wavelengths for ESIPT dyes embedded in crystalline or amorphous solids. To this end, the Adamo [107,111,112] and Crespo-Otero [113,114] groups came out with refined protocols that likely stand today as the most elegant and refined available. In what concerns the intensity of the emission in the solid-state, several studies clearly confirmed that RACI explains the strong increase of emission intensity in the more constrained solid-state [105,106,108,109,110]. Whilst these works provide quantitative estimates of the increased energetical cost of the twisting in going from the solvent to the solution, quantitative estimates of the emission quantum yield in the solid state remains typically beyond reach today. Nevertheless, one should certainly point out the very neat 2020 study of Dommett et al., who investigated a series of eleven crystals containing ESIPT dyes and obtained impressive correlation between experimental and computed data, allowing them to propose design rules for ESIPT emitters in the solid state [115].

## 6. Conclusions

This article briefly describes the database of the so far reported organic fluorophores which show (1) a full or partial ESIPT emission due to the presence of an intramolecular H-bond in their structure and (2) fluorescence intensity (QY > 10%) both in solution and in the solid-state. The variety of chromophores reported is already important but many involve benzazole or oxazole rings as proton acceptor and phenol as proton donor. There is no doubt that in the near future, many more elegant examples will be added to this attractive class of compounds. Within the global context of sustainable development, it is of upmost importance to construct accessible probes which can target as many applications as possible. DSE dyes can be simultaneously applied to various luminescent displays working in solution (sensing, imaging) or as solids (optoelectronic devices, inks) and therefore appear as bright alternatives to current luminescent probes.

## Figures and Tables

**Figure 1 molecules-27-02443-f001:**
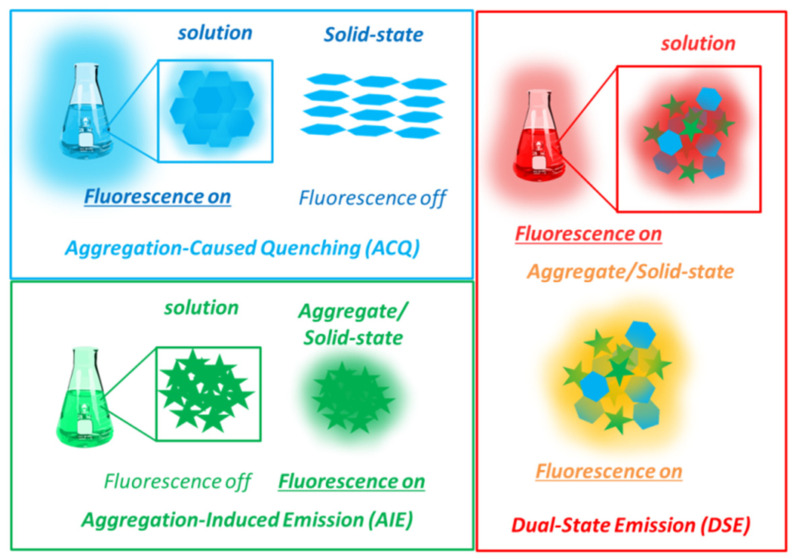
Schematic representations of the concepts of aggregation-caused quenching (ACQ), aggregation-induced emission (AIE) and dual-state emission (DSE).

**Figure 2 molecules-27-02443-f002:**
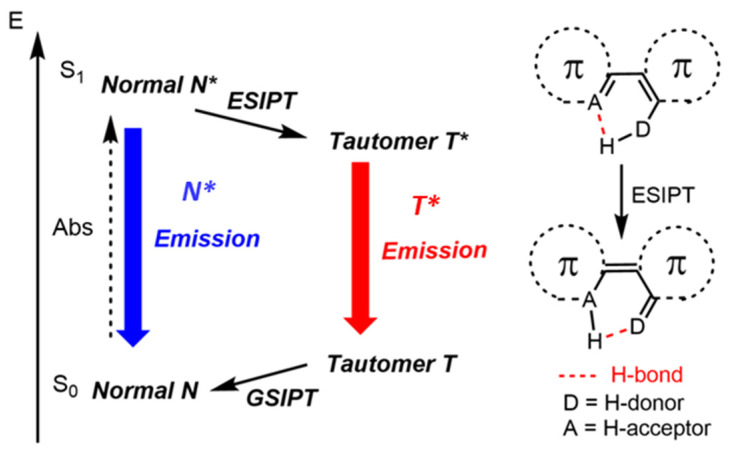
Schematic representation of the four-level phototautomerization process of ESIPT. * represents the excited species.

**Figure 6 molecules-27-02443-f006:**
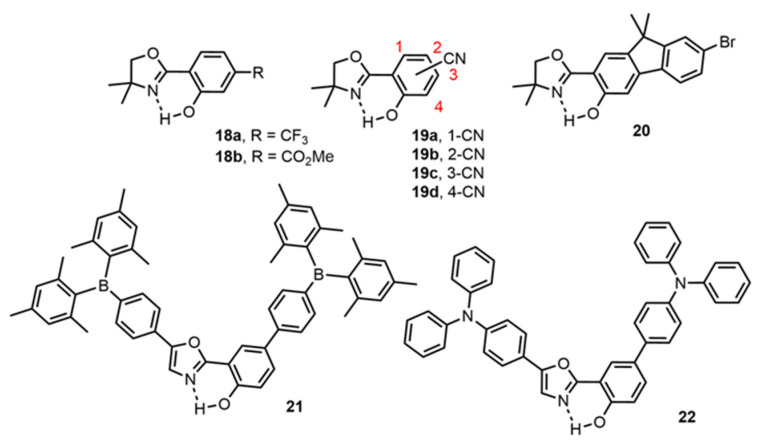
DSE/ESIPT fluorophores based on an oxazole scaffold [72,79,80,81].

**Figure 7 molecules-27-02443-f007:**
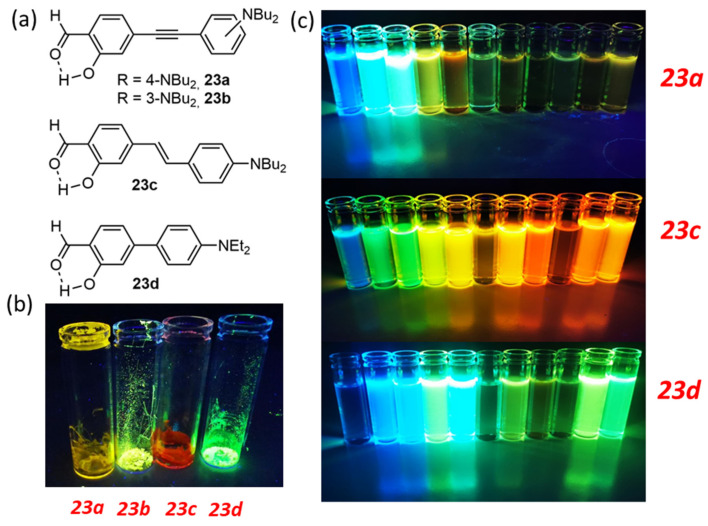
(**a**) DSE/ESIPT fluorophores based on a salicylaldehyde scaffold and photographs of dyes **23a–d** under irradiation (λ_exc_ = 365 nm) in (**b**) solution (from left to right: cyclohexane, toluene, diethylether, THF, dichloromethane, protonated dichloromethane, acetone, acetonitrile, ethanol, DMSO and DMF) and (**c**) in the solid-state as powders [82]. Adapted from ref [82] Copyright 2021 John Wiley and Sons.

**Figure 8 molecules-27-02443-f008:**
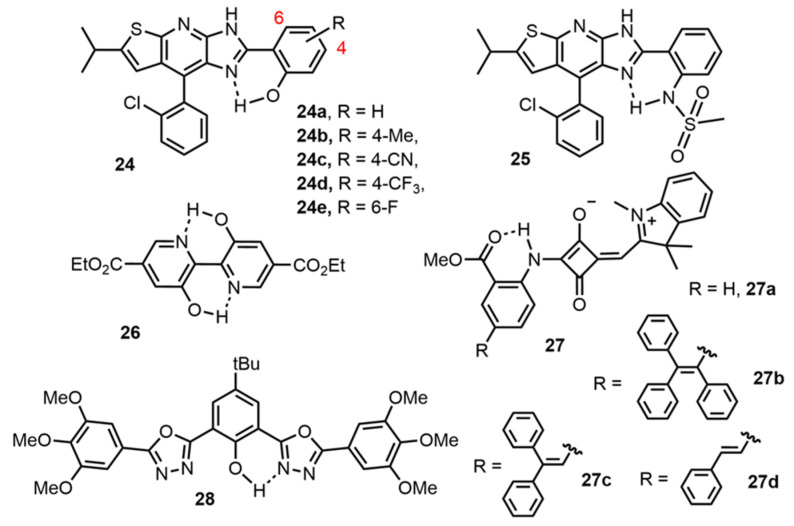
Examples of DSE/ESIPT fluorophores based on miscellaneous scaffolds [83,84,85,86].

## Data Availability

Not applicable.

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
