# Peer review of "Excited-State Intramolecular Proton Transfer Dyes with Dual-State Emission Properties: Concept, Examples and Applications"

_molecules, 2022, doi:10.3390/molecules27082443_

Round 1

Reviewer 1 Report

In this manuscript, the authors introduced the concept of dual-state emissive (DSE) dyes and the current progress of their development. This review covers majority of the DSE fluorophores which exhibits high fluorescence quantum yield in both solution and solid states. Overall, this paper offers a timely review of the DSE dyes and is a great contribution to the field. There are a few questions/comments the authors need to address in a revised version.

1. Please give an example of phenol (E*) and keto (K*) forms in Figure 2.

2. Line 191, the authors mentioned “chart 1”, should it be “table 1”?

3. In the paper, the authors tried to use the concept of restricted access to conical intersection (RACI) to explain the low quantum yield of DSE dye in solution. Is the formation of CI a competing process of ESIPT? Or is it a process that follows the ESIPT? In other words, does CI inhibit ESIPT or does CI inhibit the fluorescence of ESIPT product?

4. In solid state, most DSE dyes only emit after ESIPT, so they typically only have one emission peak. Please explain the dual-color emission of Dye 14 in table 1 in the manuscript. This is an interesting phenomenon.

5. In solution, the dye can emit fluorescence before and after ESIPT, right? The emission before ESIPT is typically bluer than it after ESIPT. Is that why some of the dyes have a dual-color emission in the solution state? For example, dye 5, 6a, 10, 11b. The authors should discuss this in the paper.

6. The dyes with electron donating or accepting groups can introduce intermolecular charge transfer (ICT) process on the excited state, could the authors discuss the relationship between ICT and ESIPT in these dyes?

Author Response

List of changes and reply to the comments

Reviewer #1: In this manuscript, the authors introduced the concept of dual-state emissive (DSE) dyes and the current progress of their development. This review covers majority of the DSE fluorophores which exhibits high fluorescence quantum yield in both solution and solid states. Overall, this paper offers a timely review of the DSE dyes and is a great contribution to the field. There are a few questions/comments the authors need to address in a revised version.

[1] Please give an example of phenol (E*) and keto (K*) forms in Figure 2.

Reply: On figure 2, we decided to replace E* and K* by N* and T* on the transition arrows to widen the scope of examples of probes displaying ESIPT emission. Moreover, in the text, figure 2 clearly refers to “photoinduced tautomerization between an excited normal (N*) and a tautomeric species (T*)” which is consistent with the modification. As such, we feel there is no point anymore in giving an example of E* and K* at this stage.

[2] Line 191, the authors mentioned “chart 1”, should it be “table 1”?

Reply: We apologize for the mistake. It should be “figure 3”. It has been modified.

[3] In the paper, the authors tried to use the concept of restricted access to conical intersection (RACI) to explain the low quantum yield of DSE dye in solution. Is the formation of CI a competing process of ESIPT? Or is it a process that follows the ESIPT? In other words, does CI inhibit ESIPT or does CI inhibit the fluorescence of ESIPT product?

Reply: ESIPT process is extremely fast, usually in the sub-picosecond timescale, which is far less than usual conical intersections (CI) dynamics. However, ESIPT fluorescence emission is in the nanosecond timescale. In other words, ESIPT occurs far before CI which can in turn stabilize a dark state and significantly quench fluorescence. The RACI concept rationalizes a least accessible CI state by appropriate electronic effects. For more information on RACI, the reader is invited to read reference 58.

[4] In solid state, most DSE dyes only emit after ESIPT, so they typically only have one emission peak. Please explain the dual-color emission of Dye 14 in table 1 in the manuscript. This is an interesting phenomenon.

Reply: What is particularly interesting for dye 14 is that a full frustration of ESIPT in solution turns into a partial restoration of the ESIPT in the solid-state. The authors state that “The dual emission in the solid state for ESIPT compound could be due to restriction of cis-trans transformation of the keto form in the excited state due to tight molecular packing. In the solid state, an intramolecular hydrogen bond dominates over intermolecular hydrogen bonding due to physical constraints resulting into an ESIPT process”. A sentence has been added in the manuscript.

[5] In solution, the dye can emit fluorescence before and after ESIPT, right? The emission before ESIPT is typically bluer than it after ESIPT. Is that why some of the dyes have a dual-color emission in the solution state? For example, dye 5, 6a, 10, 11b. The authors should discuss this in the paper.

Reply: The dual E*/K* emission is highly dependent on the proticity of the solvent and intrinsic electronic effects. Protic solvents such as alcohols tend to stabilize the E* level, leading to a proton back-transfer after the occurrence of ESIPT. A sentence has been added in the text to explain this phenomenon.

[6] The dyes with electron donating or accepting groups can introduce intermolecular charge transfer (ICT) process on the excited state, could the authors discuss the relationship between ICT and ESIPT in these dyes?

Reply: ESIPT should be only seen as excited-state dynamics between two distinct singlet states, i.e. two excited species which can interact with the environment depending on their own physical properties. Upon introduction of electro-donating or -accepting groups, a dipole moment is created on the main molecular backbone, leading to the stabilization of an ICT state, whose energy is highly influenced by the nature of the solvent in solution. Therefore, it is not possible to establish a relationship between ESIPT and ICT, as the latter differs within each molecule where ESIPT can take place.

Reviewer 2 Report

The MS “Excited-State Intramolecular Proton Transfer Dyes with Dual-State Emission Properties: Concept, Examples and Applications” by T. Stoerkler, T. Pariat, A.D. Laurent, D. Jacquemin, G. Ulrich and J. Massue is a mini-review, which is dealing with organic compounds, demonstrating the excited state intramolecular proton transfer reaction (ESIPT) and intense fluorescence both in the solid (crystalline) state and in solutions in various organic solvents.

The subject of this paper is interesting and important for many practical applications, thus it worth publication in Molecules.

However, there are several moments, which needs additional clarification or revision, which generally will improve the science level of this MS.

First confusing circumstance is in the title and main idea of this paper: “Dual-State Emission” according to the Authors refers to fluorescence in two different aggregation states, solid and dissolved. At the same time, ESIPT compounds are also “double/multiple excited states” emitters. Fluorescence from the Normal and Tautomer species (and sometimes – from Anionic ones as well) are involved in ESIPT phenomenon. Thus, I would like to propose to modify the Authors “Dual-State” definition to “Dual Aggregation State” and correspondently, the abbreviation DSE to DASE all through the text of the MS (DSE is used 48 times, Dual-state - 15). Alternatively, “Multiple Aggregation State Emission” (MASE) can be proposed…

Line 79 (and also 100, 146, 263, 398): “sizable emission”… I am not a natively English-speaking person, probably, this is fully correct. However, reading science literature on molecular fluorescence from ~1981, I do not remember such a words combination used anywhere. Is it possible to change it to “intense fluorescence”, “bright fluorescence”, “high QY fluorescence” or something else?

Line 83: the Authors discuss the aggregation-caused quenching (ACQ) in several classes of organic molecules. This is not fully correct, because many of the listed compounds form EXCIMERs, which are characterized by more intense emission in respect to their monomeric form. The most known and classic example is pyrene. This text fragment needs corresponding correction.

Line 101: the Authors characterize “DSE”-compounds: “These dyes have been far less studied”… If one refer, for example, to the classic monograph on molecular fluorescence by C.A. Parker (issued in ~1968), he/she can find many examples of organic compounds demonstrated intense emission both in solid and dissolved states.

Lines 124-125: “excited state dynamics” cannot prevent π-stacking, because the latter takes place first of all in the ground state…

Lines 145-146: “limited but growing scope of ESIPT emitters displaying sizable emission intensity in multiple environments” – if ESIPT compounds are fluorescent in fluid solutions, its intense solid state emission is a GENERAL rule. Such molecules are much more fluorescent in confined (rigid) surrounding.

Line 188: the Authors declare, that benzimidazolic ESIPT-compounds are characterized by “reducing non-radiative desexcitations”, but present no clarification (science background) of this statement…

Structural formula presentation in Figures 3 and 4 is not fully successful. Can the Authors use R1, R2, etc. instead of their “⸾–≡–…”? Here I am trying to reproduce the Authors’ graphics in ASCII mode, if it is generally possible).

Lines 216-217: “pyridinium was able to stabilize the K* state by resonance and therefore DISFAVOR NON-RADIATIVE DEACTIVATIONS, leading to intense fluorescence in solution” – please, explain, how… What is the correlation between resonance (intramolecular interactions) and non-radiative deactivation in this concrete case?

Line 221: “strong quenching of emission in the solid-state, presumably owing to UNFAVORABLE ELECTROSTATIC INTERACTIONS” – the same, please explain the mechanism of the influence of electrostatic interactions in this concrete case.

Line 239: In many cases, Authors of the cited original paper introduce silyl moieties. With what aim? What is the role of -SiR3 groups on fluorescence in solid and dissolved states?

Line 254: “hypochlorous acid HCOCl” – the latter is not a hypochlorous acid…

Line 268: capture to Figure 6. Compounds 21 and 22 contain OXAZOLE, not “OXAZOLINYL” moieties…

Lines 285-292 and later discussion: “simple salicylaldehyde derivatives 23a-d”. First, they are not SIMPLE. Second, the readers would like to see their absorption and fluorescence spectra to conclude, whether ESIPT was generally observed in the case of the above-mentioned compounds or not. I suspect existence of only solvent polarity induced spectral shifts for these compounds and no ESIPT at all… Presented photographs is not enough in such a case.

Figure 8, compound 27 – is it generally capable to ESIPT? I am not sure in realization of a TICT mechanism in this case as well…

Line 323: what does it mean “solvated medium”? Medium, solvated by what?

General remark for Section 3. There are no mentioning of a wide and very popular during the last decades class of ESIPT fluorophores – derivatives and analogs of 3-hydroxy-flavone (3-hydroxy-2-aryl/hetaryl-chromones). All of them are characterized by efficient ESIPT process, intermediate-to-high fluorescence QYs, high Stokes shift and intense fluorescence both in solutions and in crystalline/solid state. To my understanding, without discussion of 3-hydroxy-chromones this review is SIGNIFICANTLY INCOMPLETE…

Section 4. I would like to recommend the Authors to add data of absorption spectra (at least, positions of the long-wavelength absorption bands maxima).

Section 5 looks low-informative. Difficulties in modeling of fluorescent properties in the crystalline state are significant and intuitively understandable to everyone… The existence of a general mechanism of radiationless decay for any ESIPT compound was known from 1987 (S-I Nagaoka, 10.1016/0047-2670(87)87054-5), however, its nature still awaits for elucidation. This is not a classical TICT, because it is typical to many ESIPT compounds, for which TICT is not possible principally.

Basing on all the above considerations, I would like to propose MAJOR REVISION of this MS before its future acceptance by Molecules.

Author Response

Reviewer #2: The MS “Excited-State Intramolecular Proton Transfer Dyes with Dual-State Emission Properties: Concept, Examples and Applications” by T. Stoerkler, T. Pariat, A.D. Laurent, D. Jacquemin, G. Ulrich and J. Massue is a mini-review, which is dealing with organic compounds, demonstrating the excited state intramolecular proton transfer reaction (ESIPT) and intense fluorescence both in the solid (crystalline) state and in solutions in various organic solvents.

The subject of this paper is interesting and important for many practical applications, thus it worth publication in Molecules. However, there are several moments, which needs additional clarification or revision, which generally will improve the science level of this MS.

[1] First confusing circumstance is in the title and main idea of this paper: “Dual-State Emission” according to the Authors refers to fluorescence in two different aggregation states, solid and dissolved. At the same time, ESIPT compounds are also “double/multiple excited states” emitters. Fluorescence from the Normal and Tautomer species (and sometimes – from Anionic ones as well) are involved in ESIPT phenomenon. Thus, I would like to propose to modify the Authors “Dual-State” definition to “Dual Aggregation State” and correspondently, the abbreviation DSE to DASE all through the text of the MS (DSE is used 48 times, Dual-state - 15). Alternatively, “Multiple Aggregation State Emission” (MASE) can be proposed…

Reply: We beg to disagree with the reviewer on that special point. Not all ESIPT dyes reviewed in this account display aggregation behavior in solution. In the majority of cases, only the emission of the molecular species can be detected using common spectroscopic measurements. Moreover, one should distinguish between “Dual-state emission”, i.e. solution- and solid-state and “Dual emission”, i.e. simultaneous emission from two distinct excited states, e.g. N*/T* or E*/K*. To justify these abbreviation, we referred to two recent reviews on the subject by Rodriguez-Molina who coined the term dual-state emission “Dual-​State Emission (DSE) in Organic Fluorophores: Design and Applications” and “One molecule to light it all: The era of dual-​state emission” (references 34 and 35 in the manuscript). For consistency with the literature, we feel like it is better to keep our current title.

[2] Line 79 (and also 100, 146, 263, 398): “sizable emission”… I am not a natively English-speaking person, probably, this is fully correct. However, reading science literature on molecular fluorescence from ~1981, I do not remember such a words combination used anywhere. Is it possible to change it to “intense fluorescence”, “bright fluorescence”, “high QY fluorescence” or something else?

Reply: The reviewer is right. We have changed everywhere throughout the manuscript “sizable emission” for “intense fluorescence emission” as requested.

[3] Line 83: the Authors discuss the aggregation-caused quenching (ACQ) in several classes of organic molecules. This is not fully correct, because many of the listed compounds form EXCIMERs, which are characterized by more intense emission in respect to their monomeric form. The most known and classic example is pyrene. This text fragment needs corresponding correction.

Reply: A sentence underlining the possible formation of excimers in pyrene derivatives has been added in the text.

[4] Line 101: the Authors characterize “DSE”-compounds: “These dyes have been far less studied”… If one refer, for example, to the classic monograph on molecular fluorescence by C.A. Parker (issued in ~1968), he/she can find many examples of organic compounds demonstrated intense emission both in solid and dissolved states.

Reply: The scope of this review has been clearly established in the introduction: an overview of organic dyes displaying 1) full or partially frustrated ESIPT process and 2) reported photophysical properties, studied in solution and in the solid-state showing fluorescence quantum yields reaching at least 10% in these two media. Nevertheless, we are completely aware that many dyes were reported in the literature at times where spectroscopic measurements did not allow accurate recording of luminescence intensity.

[5] Lines 124-125: “excited state dynamics” cannot prevent π-stacking, because the latter takes place first of all in the ground state…

Reply: Not necessarily. Although it is completely true that a lot of stacking processes take place in the ground-state, excited-state dynamics such as excimer emission can also be seen as a form of excited π-stacking.

[6] Lines 145-146: “limited but growing scope of ESIPT emitters displaying sizable emission intensity in multiple environments” – if ESIPT compounds are fluorescent in fluid solutions, its intense solid state emission is a GENERAL rule. Such molecules are much more fluorescent in confined (rigid) surrounding.

Reply: What we meant is that the majority of ESIPT dyes are heavily quenched in solution but emissive in solid (general and most encountered case). Other examples are now present in the literature: 1) ESIPT dyes presenting emission in solution and in solid (the scope of this review), 2) ESIPT dyes presenting only emission in solution and not in solid (rarely encountered. See ref 69 for scope and examples).

[7] Line 188: the Authors declare, that benzimidazolic ESIPT-compounds are characterized by “reducing non-radiative desexcitations”, but present no clarification (science background) of this statement

Reply: The reader/reviewer should find scientific clarification on that topic in ref. 64, at the end of the manuscript. Briefly, the authors explain that the electronic nature of nitrogen disfavor the formation of a twisted ICT process which is much more favorable in HBO/HBT.

[8] Structural formula presentation in Figures 3 and 4 is not fully successful. Can the Authors use R1, R2, etc. instead of their “⸾–≡–…”? Here I am trying to reproduce the Authors’ graphics in ASCII mode, if it is generally possible).

Reply: This has been changed.

[9] Lines 216-217: “pyridinium was able to stabilize the K* state by resonance and therefore DISFAVOR NON-RADIATIVE DEACTIVATIONS, leading to intense fluorescence in solution” – please, explain, how… What is the correlation between resonance (intramolecular interactions) and non-radiative deactivation in this concrete case?

Line 221: “strong quenching of emission in the solid-state, presumably owing to UNFAVORABLE ELECTROSTATIC INTERACTIONS” – the same, please explain the mechanism of the influence of electrostatic interactions in this concrete case.

Reply: This account aims at reviewing literature on a given topic (here DSE dyes showing ESIPT), not providing a deep and complete explanation on every peer-reviewed publication quoted in the reference section. For more information on the above issues, please read reference 69.

[10] Line 239: In many cases, Authors of the cited original paper introduce silyl moieties. With what aim? What is the role of -SiR3 groups on fluorescence in solid and dissolved states?

Reply: The introduction of silyl moieties has only a solubility advantage, along with a stabilization effect but no electronic influence on the photophysical properties. Please check references 67 and 68 for more details.

[12] Line 254: “hypochlorous acid HCOCl” – the latter is not a hypochlorous acid…

Reply: It has been changed to just “HCOCl”.

[13] Line 268: capture to Figure 6. Compounds 21 and 22 contain OXAZOLE, not “OXAZOLINYL” moieties…

Reply: We used the same term “oxazolinyl”, as that employed by the authors of the related publications (refs 77 and 78). Oxazole has been added on the legend of figure 6.

[14] Lines 285-292 and later discussion: “simple salicylaldehyde derivatives 23a-d”. First, they are not SIMPLE. Second, the readers would like to see their absorption and fluorescence spectra to conclude, whether ESIPT was generally observed in the case of the above-mentioned compounds or not. I suspect existence of only solvent polarity induced spectral shifts for these compounds and no ESIPT at all… Presented photographs is not enough in such a case.

Reply: Again, we just wish to review the optical properties of these peer-reviewed reported compounds, as described in reference 80 where the reviewer will find more information on the photophysics of the related dyes. The word “simple” has been removed from this manuscript.

[15] Figure 8, compound 27 – is it generally capable to ESIPT? I am not sure in realization of a TICT mechanism in this case as well…

Reply: The reviewer is advised to check reference 83 for more details on this compound.

[16] Line 323: what does it mean “solvated medium”? Medium, solvated by what?

Reply: We changed “solvated medium” by “solvent”.

[17] General remark for Section 3. There are no mentioning of a wide and very popular during the last decades class of ESIPT fluorophores – derivatives and analogs of 3-hydroxy-flavone (3-hydroxy-2-aryl/hetaryl-chromones). All of them are characterized by efficient ESIPT process, intermediate-to-high fluorescence QYs, high Stokes shift and intense fluorescence both in solutions and in crystalline/solid state. To my understanding, without discussion of 3-hydroxy-chromones this review is SIGNIFICANTLY INCOMPLETE…

Reply: We feel like this falls out of the scope of this review which is accounting for DSE dyes presenting ESIPT emission and optical studies in solution and solid-state with QY reaching at least 10%. 3-hydroxyfavone are ESIPT dyes which have been indeed widely studied for their photophysical properties but most of the publications date back from a decade ago or more where solid-state spectroscopy did not allow calculations of accurate quantum yields. Alternatively, some recent papers only focus on optical properties in solution and related photophysics. Nevertheless, we have included a remark about these dyes and three references at the end of section 4.

[18] Section 4. I would like to recommend the Authors to add data of absorption spectra (at least, positions of the long-wavelength absorption bands maxima).

Reply: The absorption data have been added to table 1.

[19] Section 5 looks low-informative. Difficulties in modeling of fluorescent properties in the crystalline state are significant and intuitively understandable to everyone… The existence of a general mechanism of radiationless decay for any ESIPT compound was known from 1987 (S-I Nagaoka, 10.1016/0047-2670(87)87054-5), however, its nature still awaits for elucidation. This is not a classical TICT, because it is typical to many ESIPT compounds, for which TICT is not possible principally.

Reply: The reviewer is kindly invited to read the various references listed in this section.

Round 2

Reviewer 2 Report

Most of my remarks were taken into account or at least answered.

I still cannot agree with the Authors main definition DSE, “dual state emission”. The reader would hardly distinguish it from “dual band emission” before reading this Review.. However, if this not the first time of its usage in the science literature, let it remain “as it is” in the paper.

The thing, this MUST be corrected – line 257, HCOCl. Please re-read once more the ref.75, at least, its Abstract ”…logic-based visualization of thiophenol (stressor) and HOCl (thiophenol-activated stress response product) in vivo

Comment on my remark [19] (The reviewer is kindly invited to read the various references listed in this section.) looks a bit caddish… However, this has no connection to the general evaluation of this Review.

Resuming, accept after compulsory correction of “HCOCl” to HOCl (line 257 of a version 2)…

Author Response

"HCOCl" has been corrected to "HOCl". We apologize for this typo and acknowledge again the reviewer for his/her thorough evaluation of our manuscript